# Impact of High-to-Moderate Penetrance Genes on Genetic Testing: Looking over Breast Cancer

**DOI:** 10.3390/genes14081530

**Published:** 2023-07-26

**Authors:** Antonella Turchiano, Marilidia Piglionica, Stefania Martino, Rosanna Bagnulo, Antonella Garganese, Annunziata De Luisi, Stefania Chirulli, Matteo Iacoviello, Michele Stasi, Ornella Tabaku, Eleonora Meneleo, Martina Capurso, Silvia Crocetta, Simone Lattarulo, Yevheniia Krylovska, Patrizia Lastella, Cinzia Forleo, Alessandro Stella, Nenad Bukvic, Cristiano Simone, Nicoletta Resta

**Affiliations:** 1Medical Genetic, Department of Precision and Regenerative Medicine and Ionian Area (DiMePRe-J), University of Bari “Aldo Moro”, 70124 Bari, Italy; antonella.turchiano@uniba.it (A.T.); marilidia.piglionica@uniba.it (M.P.); s.martino5@studenti.uniba.it (S.M.); rosanna.bagnulo@uniba.it (R.B.); a.garganese80@gmail.com (A.G.); annunziata.deluisi@uniba.it (A.D.L.); stefania.chirulli@gmail.com (S.C.); m.iacoviello3@gmail.com (M.I.); michele.stasidoc@libero.it (M.S.); nelatabaku96@gmail.com (O.T.); eleonora.meneleo@gmail.com (E.M.); martycapurso@gmail.com (M.C.); silviacrocetta96@gmail.com (S.C.); lattarulo42@gmail.com (S.L.); krylovskayevheniia@gmail.com (Y.K.); alessandro.stella@uniba.it (A.S.); nenadbukvic@virgilio.it (N.B.); cristiano.simone@uniba.it (C.S.); 2Rare Disease Center, Internal Medicine Unit “C. Frugoni”, AOU Policlinico di Bari, 70124 Bari, Italy; patrizia.lastella76@gmail.com; 3Cardiology Unit, Department of Precision and Regenerative Medicine and Ionian Area (DiMePRe-J), University of Bari “Aldo Moro”, 70124 Bari, Italy; cinzia.forleo@uniba.it; 4Medical Genetics, National Institute of Gastroenterology, “S. de Bellis” Research Hospital, Via Turi 27, Castellana Grotte, 70013 Bari, Italy

**Keywords:** breast cancer, moderate-penetrance genes, *CHEK2*, *ATM*, *PALB2*, *RAD51C*, *RAD51D*

## Abstract

Breast cancer (BC) is the most common cancer and the leading cause of cancer death in women worldwide. Since the discovery of the highly penetrant susceptibility genes *BRCA1* and *BRCA2*, many other predisposition genes that confer a moderate risk of BC have been identified. Advances in multigene panel testing have allowed the simultaneous sequencing of *BRCA1/2* with these genes in a cost-effective way. Germline DNA from 521 cases with BC fulfilling diagnostic criteria for hereditary BC were screened with multigene NGS testing. Pathogenic (PVs) and likely pathogenic (LPVs) variants in moderate penetrance genes were identified in 15 out of 521 patients (2.9%), including 2 missense, 7 non-sense, 1 indel, and 3 splice variants, as well as two different exon deletions, as follows: *ATM* (*n* = 4), *CHEK2* (*n* = 5), *PALB2* (*n* = 2), *RAD51C* (*n* = 1), and *RAD51D* (*n* = 3). Moreover, the segregation analysis of PVs and LPVs into first-degree relatives allowed the detection of *CHEK2* variant carriers diagnosed with in situ melanoma and clear cell renal cell carcinoma (ccRCC), respectively. Extended testing beyond *BRCA1/2* identified PVs and LPVs in a further 2.9% of BC patients. In conclusion, panel testing yields more accurate genetic information for appropriate counselling, risk management, and preventive options than assessing *BRCA1/2* alone.

## 1. Introduction

Breast cancer (BC) is the most common diagnosed cancer and the leading cause of cancer death in women worldwide [1]. A combination of genetic and nongenetic factors affects BC occurrence: about 5–10% of all cases of BC are related to genetic predisposition or family history, and the remaining 90–95% of cases are connected to environmental factors and lifestyle.

*BRCA1* and *BRCA2* are the genes most commonly associated with hereditary BC, whose proteins are involved in the double-strand DNA damage reparation mechanism [2]. With an average cumulative lifetime risk of about 70% for *BRCA1/2* carriers [3], genetic testing is clinically indicated in individuals with hereditary BC [4]. Despite *BRCA1* and *BRCA2* being the two high-penetrance genes mainly correlated with increased risk of hereditary BC, many other predisposition genes have been discovered [5]. Mutations in the *TP53*, *PTEN*, *STK11*, and *CDH1* genes, which are known to play a role in well-described syndromes (Li-Fraumeni syndrome, Cowden syndrome, Peutz-Jeghers syndrome, and hereditary diffuse gastric cancer syndrome) confer an increased risk of BC [5,6]. Furthermore, the role of moderate-risk susceptibility genes in BC predisposition is constantly emerging. Among those genes, *PALB2*, *CHEK2*, *ATM*, *RAD51C*, and *RAD51D* encode proteins involved in BRCA protein stability and/or function which altogether form a part of the homologous recombination (HR) DNA repair pathway [2]. BC risk increases to 15–35% in moderate-risk susceptibility gene carriers, and the NCCN recommends annual mammogram and breast magnetic resonance imaging (MRI) for all these individuals [4,7]. Conversely, insufficient evidence has been collected so far to establish a risk reduction option, except for PALB2 mutation carriers [4].

In the era of multigene panel testing, next-generation sequencing (NGS) provides a unique platform to study several genes in a short time in a cost-effective way and in a huge number of patients [8]. This approach yields more accurate genetic information for appropriate counselling and risk management for patients with PVs and LPVs in other genes than assessing *BRCA1/2* alone, and may lead to improve therapeutic and preventive options [9]. In addition, family members at high risk for cancer may be eligible for genetic testing and medical interventions.

In this study, the data of 521 BC cases who underwent a multigene NGS test were reviewed to assess the mutation occurrences in genes other than *BRCA1/2* and to define a common clinicopathologic profile. Finally, a segregation analysis of detected variants in first-degree family members was performed.

## 2. Materials and Methods

### 2.1. Patients’ Recruitment

A group of 521 patients diagnosed with BC were referred for diagnostic purposes to the Medical Genetic Unit at the University of Bari from January 2018 to April 2023. Since they fulfilled the criteria concerning cancer personal and family history according to the NCNN guideline, genetic testing was proposed [4]. Posttest counselling was offered to patients carrying genetic variants, and segregation analysis of PVs and LPVs into first-degree relatives was performed. All the individuals provided informed consent for the study.

### 2.2. Next-Generation Sequencing (NGS)

Peripheral blood (PB) was collected from all the patients. Genomic DNA extraction was performed using a QIAamp Mini Kit (Qiagen, Hilden, Germany), following the manufacturer’s instructions. Its concentration and quality were evaluated using a Qubit ds DNA HS Assay Kit on a Qubit 2.0 Fluorimeter (Invitrogen, Carlsbad, CA, USA), according to the manufacturer’s instructions. Two multigene panels were used to test the genes involved in cancer predisposition. In detail, the genomic DNA of patients enrolled from January 2018 to May 2022 was investigated with a custom Illumina AmpliSeq Panel (Illumina, San Diego, CA, USA) which included the coding sequencing (CDS) and ±25 bp of intronic flanking regions of 23 genes associated with familial cancer (*EPCAM*, *TP53*, *RAD51D*, *RAD51C*, *RAD50*, *PTEN*, *ATM*, *MSH2*, *MSH6*, *CHEK2*, *STK11*, *APC*, *MLH1*, *MUTYH*, *PALB2*, *BARD1*, *CDH1*, *BRIP1*, *PMS2*, *XRCC2*, *NBN*, *BRCA1*, and *BRCA2*). The panel was designed online using the Design Studio tool provided by Illumina (designstudio.illumina.com (accessed on 11 November 2020)). Library preparation was carried out by following the Illumina Ampliseq workflow (Illumina, San Diego, CA, USA). Sequencing runs were conducted on an Illumina Miseq instrument (Illumina, San Diego, CA, USA) following the manufacturer’s instructions using a standard flow cell (Illumina, San Diego, CA, USA) and a V2 300 cycle cartridge (Illumina, San Diego, CA, USA). Conversely, for the remaining individuals of the cohort, the TruSight Hereditary Cancer Panel provided by Illumina (Illumina, San Diego, CA, USA), including 113 genes related to cancer predisposition, was utilized. The panel comprehended the CDSs (±20 bp of intronic flanking regions) of the selected 113 genes. The Nextera Flex for Enrichment protocol (Illumina, San Diego, CA, USA) was performed to prepare the libraries. A NextSeq instrument (Illumina, San Diego, CA, USA) was used to perform the sequencing run by using a mid-output flow cell and cartridge.

### 2.3. Genetic Variant Classification

Data analysis was performed using the Miseq control software (Illumina, San Diego, CA, USA) and the NextSeq control software (Illumina, San Diego, CA, USA), respectively. The BAM were generated using the BWA Aligner software [10,11]. The variant calling was performed using the Genome Analysis Toolkit (GATK) software [12]. For the former panel, a mean read per sample of about 800,000 with a mean read length of 150 was generated. The mean coverage was about 200× with a uniformity of base coverage of about 95%. The latter panel sequencing produced a mean read per patient of about 4,000,000 with a mean read length of 150. The average base coverage depth was about 800× with a uniformity of base coverage of 98.8%. To visually inspect the BAM file, an ALAMUT Visual Plus Genome Viewer (Sophia Genetics, Lausanne, Switzerland) was used. The retained variants were categorized into pathogenic (PVs) or likely pathogenic variants (LPVs) by means of several online databases (e.g., ClinVar, LOVD, Varsome, Franklin) and according to the ACMG criteria [13]. All the detected variants were described in accordance with the Human Genome Variants Society (HGVS) recommendations [14].

### 2.4. Sanger Sequencing

The detected PVs and LPVs were confirmed and segregated by Sanger sequencing on a SeqStudio Genetic Analyzer instrument (Applied Biosystems, Waltham, MA, USA) according to the manufacturer’s instructions.

### 2.5. Copy Number Variants (CNVs) Detection

Fastq data generated from the Trusight Hereditary Cancer panel were analyzed with the DRAGEN Enrichment software v4.0.3 in order to detect the CNVs. All the deleterious CNVs identified were confirmed by multiplex ligation-dependent probe amplification (MLPA) MRC Holland^®^ (Amsterdam, The Netherlands). In detail, SALSA MLPA Probemixes P045 *BRCA2/CHEK2*- D1-0519 (MRC-Holland, Amsterdam, The Netherlands) and P041 *ATM*-1-B1-0220 (MRC-Holland, Amsterdam, The Netherlands) were used. The MLPA data were analyzed by Coffalyser.Net-MRC Holland software^®^ (Amsterdam, The Netherlands).

## 3. Results

### 3.1. Detection of PVs and LPVs in BC Moderate Penetrance Genes

Germline DNA from 521 patients diagnosed with BC were retrospectively analyzed to assess PV and LPV frequency in cancer susceptibility genes other than *BRCA1/2*. The analysis revealed 15 out of 521 (2.9%) individuals harboring PVs/LPVs in five BC predisposing genes (*PALB2*, *CHEK2*, *ATM*, *RAD51C*, and *RAD51D*). In detail, 4 individuals (0.76%) were shown to carry PVs and LPVs in the *ATM* gene, 5 individuals (0.95%) in the *CHEK2* gene, 2 individuals (0.38%) in the *PALB2* gene, and 3 individuals (0.58%) in the *RAD51D* gene, and 1 PV was disclosed in the *RAD51C* gene (0.19%). The distribution of the PVs/LPVs is showed in Figure 1.

Overall, as regards the mutational impact upon protein, 2 missense variants out of 15 (13.3%), 7 (46.6%) non-sense, 1 (6.7%) indel, and 3 (20%) splicing variants were detected (Figure 1, Figure 2, Figure 3 and Figure 4). To note, 2 samples (13.3%) showed that two microdeletions comprehended *CHEK2* exon 9 and *ATM* exon 29, respectively (Figure 2, Figure 3 and Figure 4). Overall, three variants occurring in *PALB2* (p.(Gln119*)), *ATM* (p.(Tyr2100*)), and *RAD51D* (p.(Trp268*)) were identified as recurrent in two samples each. All the variants detected are listed in Table 1.

### 3.2. Clinical and Demographic Characteristics of High-to-Moderate Penetrance Gene Positive Patients

Table 2 and Appendix A summarize the clinical and demographic characteristics of the patients, showing deleterious variants in BC susceptibility genes. The mean age at diagnosis was 46 years (ranging from 29 to 69 years). The majority of the patients (73.3%) harbored a noninvasive ductal carcinoma of no special type (IDC-NST). Only one patient developed another type of tumor beside BC (in situ melanoma), and three patients developed multiple BCs. It is noteworthy that the patient with a RAD51C PV experienced two BCs characterized by different HER2, estrogen receptor (ER), and progesterone receptor (PR) expression profiles. More than half of the selected individuals (73%) showed a family history of BC. On the other end, approximately 50% of the selected individuals displayed a family history of cancer other than BC. Unfortunately, a HER2, ER, and PR profile was not available for all the patients. Interestingly, all the patients carrying *ATM* PVs and LPVs showed the same HER2, ER, and PR status.

### 3.3. Segregation Analysis of Detected Variant in First-Degree Family Members

In agreement with the guideline, the detected PVs and LPVs were segregated into first-degree relatives. To date, segregation data are only available for six variants. The most significant results of segregation analysis are shown in Figure 5. It is noteworthy that two *CHEK2* mutations occurred in two individuals with a diagnosis of clear cell renal cell carcinoma (ccRCC) and in situ melanoma, respectively. Additionally, a pathogenic *ATM* variant was also detected in a subject who had developed BC.

## 4. Discussion

In the present study, a population of 521 patients diagnosed with BC who underwent NGS-based multigene panel testing were retrospectively analyzed. All the patients were selected according to their BC personal history and cancer family history [4]. This study aimed to assess the frequency of PVs and LPVs in BC moderate-risk cancer genes beyond *BRCA1/2*, in order to ascertain a possible correlation between deleterious variants in those genes and the genesis of cancer other than BC, and to explore the clinical, histological, and molecular BC profile. Overall, 15 patients out of 521 (2.9%) showed PVs and LPVs in five cancer-predisposing genes other than *BRCA1/2* (*PALB2*, *CHEK2*, *ATM*, *RAD51C*, and *RAD51D*). All these genes play a pivotal role in the HR mechanism [2]. In detail, the distribution of PVs and LPVs was as follows: *ATM* (0.76%), *CHEK2* (0.95%), *PALB2* (0.38%), *RAD51D* (0.58%), and *RAD51C* (0.19%). In our population study, the *RAD51C* PVs and LPVs occurrence are discovered to be in line with previous publications [7,15]. In contrast, the *RAD51D* variant frequency detected was slightly higher than previous studies, probably due to the small size of our population study [7,15]. Moreover, the frequency of the *ATM*, *PALB2*, and *CHEK2* mutations is overall lower than those reported in similar studies [16,17,18,19,20,21]. Nevertheless, many of these studies focused on *BRCA1/2* negative BC. Herein, we examined BC patients with NGS multigene panels including *BRCA1/2*, thus not excluding those carrying deleterious variants in the two high-risk genes. Among all 521 subjects, 37 showed a PVs or/and LPVs in *BRCA1/2* (7%). It is noteworthy that none of the 15 subjects displayed additional deleterious variants in other genes, making the identified PVs and LPVs the main genetic cause of BC onset.

Three variants were identified as recurrent in two patients; these were, respectively, (*PALB2*:p.(Gln119*), *ATM*:p.(Tyr2100*), and *RAD51D*:p.(Trp268*)). To our knowledge, p.(Gln119*) and p.(Tyr2100*) have been never described in the literature in individuals affected by BC. The deleterious effect of these two mutations was predicted, since *ATM* and *PALB2* loss-of-function is a known mechanism of pathogenicity, resulting in an absent or disrupted protein product. Regarding the p.(Trp268*) mutation, it has already been reported in subjects diagnosed with ovarian cancer [22,23,24], but never in BC.

Two interesting variants were detected in *CHEK2* [p.(Ile157Thr) and p.(Thr476Met)]. Both these variants were associated with attenuated BC risk [7,25,26] and have been identified in the population with a high frequency. The p.(Ile157Thr) variant also had a more favorable prognosis among BC patients [27]. This variant lies in a weakly conserved amino acid of the forkhead-associated domain, and conflicting interpretations of its pathogenicity have been published [28]. There were also discordant results on *CHEK2* protein function affected by this variant; some studies showed that p.(Ile157Thr) does not inhibit several steps in protein activation [29,30,31,32,33,34], and other studies showed disruption of the same activation [35,36]. The p.(Thr476Met) variant falls in the kinase domain of *CHEK2* protein in a moderately conserved domain. As with p.(Ile157Thr), the pathogenic classification of p.(Thr476Met) is discordant, and a previous study showed that this variant has been classified as pathogenic, likely pathogenic, and VUS by different authors [37]. By in silico analysis, the variant has been predicted to be deleterious. Kleiblova et al. (2019) showed a deleterious effect in vitro and a likely benign effect in vivo by functional studies of KAP1 phosphorylation [30]. In addition, Desrichard et al. (2011) showed reduced SOX phosphorylation equal to that of the pathogenic c.1100delC variant, suggesting that this mutation affects the ability of the recombinant *CHEK2* to recognize, bind, and phosphorylate its substrate [38].

Interestingly, within the overall mutational spectrum, two microdeletions (*CHEK2* exon 9 and *ATM* exon 29) were also identified. *CHEK2* deletions of exon 8 and of exon 9 and 10 have already been found in BC patients [39,40]. More specifically, the latter deletion is a founder mutation in the Northern and Eastern European population, accounting for a frequency of 0.9–1% in BC cases and of 0.4–0.5% in the controls, thus contributing to the BC burden [39,41]. Additionally, the *CHEK2* microdeletion was unveiled in a young patient with a strong family history of cancer. The patient’s father died of ampullary cancer, preventing us to segregate the same alterations and find a possible correlation.

As regards *ATM* exon 29 deletion, no research has been reported so far about the same deletion in cancer. Nonetheless, distinct types of *ATM* deletion have been associated with BC [42,43], also in patients revealing a BC family history. The individual in our cohort also showed a family history of BC and prostate cancer. Unfortunately, again, a segregation analysis was not available.

The contribution of the *RAD51C* and *RAD51D* deleterious variants to ovarian cancer has been previously established by numerous studies, but their role in BC predisposition is now emerging [22,44,45,46]. Several studies suggested a correlation between *RAD51C/D* PV and BC susceptibility, given their related functions in the DNA repair process via HR. Nonetheless, this remained controversial until the study of Yang et al. (2020) [47]. They estimated the relative risk of developing BC to be 1.83 for *RAD51D* PV carriers and 1.99 for *RAD51C* PV carriers, leading to the categorization of these genes as BC moderate-risk genes. Of note, one of the truncating variants [p.(Arg253*)] in the RAD51D gene identified in our population study has been previously reported in a BC patient [48]. Concerning the *RAD51C* gene, the c.1026+5_1026+7delGTA variant was identified in one patient diagnosed with BC at age 49, without a family history of cancer. Many authors classified this variant as likely pathogenic, as it impacts splicing by causing aberrant exon 8 skipping, resulting in a frame shift and a premature stop codon (p.R322Sfs22*) [49,50,51,52,53]. However, in the comprehensive splicing study of the *RAD51C* germline variants by Sanoguera-Miralles et al. (2020) [54], this alteration is reported to totally disrupt splicing, and the authors therefore proposed a pathogenic classification based on their functional study.

In addition, among all the patients tested, 5.8% had at least one variant of uncertain significance (VUS) identified in the *PALB2*, *CHEK2*, *ATM*, *RAD51C*, and *RAD51D* genes. Although there are conflicting points of view in the literature regarding the clinical management of VUS carriers, due to the unknown impact on gene function and cancer risk, the NCCN guidelines recommend screening and risk reduction strategies on the basis of personal and family history [4,55,56], which can also be evaluated by means of BC risk prediction model tools.

The segregation analysis, when available, highlighted two major results: *CHEK2* variants c.793-1G>A and p.(Ile157Thr) were found in a ccRCC case and in a subject diagnosed with in situ melanoma, respectively.

To date, the association between *CHEK2* alteration and skin cancer (both melanoma and nonmelanoma) is still unclear. Further studies are necessary to explore the involvement of *CHEK2* in the development of skin cancer. Nevertheless, *CHEK2* is reported to be a key protein in response to UV damage through its upregulation and its phosphorylation by the *ATM* and *ATR* proteins [57].

The finding of *CHEK2* PVs in a ccRCC patient is not unprecedented. Indeed, a large number of studies have been pointing out a possible connection between *CHEK2* alteration and the onset of kidney cancer (especially renal cell carcinoma) [58,59,60,61,62]. In 2004, Cybulski et al. identified an increased risk of kidney cancer associated with the p.(Ile157Thr) variant [63]. Subsequent studies with wider cohorts highlighted the peculiar occurrence of the *CHEK2* mutation in patients affected by renal cell carcinoma [58,59,60,61,62]. Additionally, Brooks et al. [64] found a *CHEK2* splicing variant in two individuals who had a diagnosis of ccRCC. Therefore, our finding adds an important piece of evidence to the literature, leading to the need of new more in-depth research to evaluate the role of *CHEK2* in renal cancer.

Regarding the clinical and demographic features of the 15 selected patients, the median age of BC onset was about 46 years. This result is approximately in line with previous reports [16,17,21,65]. Notably, Chen et al. found that *RAD51D* mutation carriers were more likely to predispose to early-onset BC, with a mean age at diagnosis of 45.4 years, similar to patients with *BRCA1* mutation (44.8 years) [66]. The mean age of 41 years in our *RAD51D* mutation carriers further confirm this finding.

Most of the cases (73.3%) exhibited an IDC-NST histology, confirming this as the most common BC histotype [67].

Furthermore, 73% and 53.3% of the cohort showed, respectively, BC and cancer other than BC in their family history. Again, these results are in line with previous studies [16,21,65].

It is noteworthy that all four subjects displaying an *ATM*-mutated BC are characterized by the same HER2, ER, and PR expression levels (HER2 negative, PR, and ER positives). This observation further confirms previous studies [68,69,70] which highlighted a strong association with an ER-positive profile [15,71]. For variants in the other four genes, we were not able to determine a common ER, PR, and HER2 molecular profile, also due to the limitation of the available histopathological data. Nevertheless, a PALB2-mutated BC exhibited a triple-negative profile, which is consistent with the literature [68,71]. Also, the *RAD51C* and *RAD51D* variants have shown a strong association with triple-negative cancer [68,70]. Here, only one case displayed a triple-negative BC. Intriguingly, a subject carrying a *RAD51D* PV developed two BCs with an opposite receptor profile. Furthermore, a link between the protein-truncating variants in *RAD51D* and ER-negative breast cancers other than the ER-positive disease has been recently defined [7,15]. In our population study, although 2 cases out of 4 confirm the subtype-specific association with PVs in *RAD51D*, the small size of our population study does not allow for strong evidence. Concerning the *CHEK2* gene, the variants have been associated with all BC histotypes except for the triple-negative [15]. Based on the limited available information, three out of five *CHEK2* PVs carriers did not show a triple-negative molecular status.

## 5. Conclusions

Given the rise in the use of multigene panel testing in diagnostic procedure, the detection of PVs and/or LPVs in a moderate-risk penetrance gene is constantly increasing. In this study, we provide estimates of PV and/or LPV occurrence in genes besides BRCA1/2 and their association either with additional forms of tumors (especially ccRCC) or with a peculiar molecular signature. Taken all together, these results may help to guide counseling, cancer screening, and other risk management strategies for women and families with PVs and/or LPVs in BC predisposition genes.

## Figures and Tables

**Figure 1 genes-14-01530-f001:**
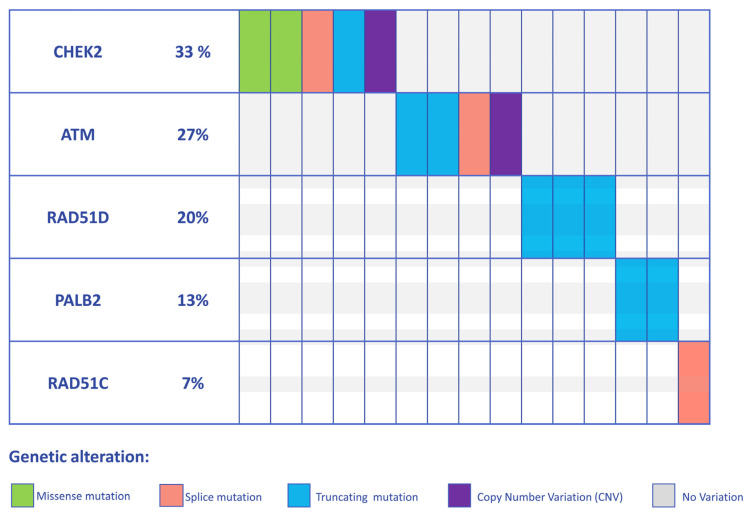
Distribution of germline PVs/LPVs in BC susceptibility genes detected with NGS multi-gene panel test in 15 BC patients. The heatmap displays the mutational spectrum identified. The graph shows both SNVs and CNVs: missense mutations, splicing mutations, and truncating mutations are represented by green, salmon pink, and light blue rectangles, respectively; exonic deletions (*CHEK2*, *ATM*) are represented by purple rectangles.

**Figure 2 genes-14-01530-f002:**
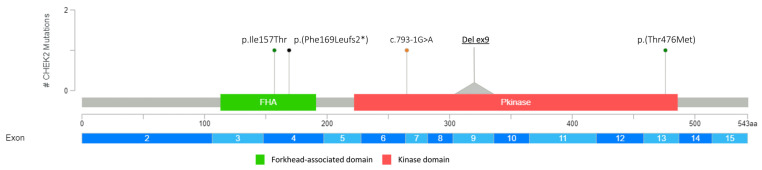
Germline PVs/LPVs identified in *CHEK2* gene. Lolliplot shows deleterious germline variants throughout the whole protein sequence of *CHEK2*. The horizontal scale bar represents the length (amino acids) of the protein sequence, and the light-blue bar below represents the corresponding exons. The vertical bar represents the number of patients affected by the specific mutation (#). Each variant identified in this study is depicted by a lolliplot. In detail, the orange lolliplot identifies the splicing mutation, the green lolliplot indicates missense variants, and the black lolliplot displays a frameshift mutation. Deletion of exon 9 is represented by the grey triangle. The lolliplot graph was obtained by the informatics tool Mutation Mapper (https://www.cbioportal.org/ (accessed on 30 May 2023) and modified.

**Figure 3 genes-14-01530-f003:**
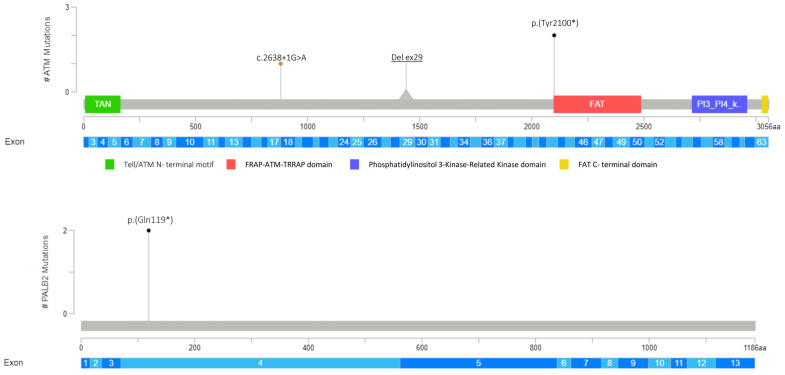
Germline PVs/LPVs identified in *ATM* and *PALB2* genes. Lolliplots show deleterious germline variants throughout the whole protein sequences of *ATM* (**above**) and *PALB2* (**below**). The horizontal scale bar represents the length (amino acids) of the protein sequence, and the light-blue bar below represents the corresponding exons. The vertical bar represents the number of patients affected by the specific mutation (#). Each variant identified in this study is depicted by a lolliplot. In detail, the orange lolliplot identifies the splicing mutation, and the black ones display non-sense mutations. Deletion of exon 29 in *ATM* is represented by the grey triangle. The lolliplot graph was obtained by the informatics tool Mutation Mapper (https://www.cbioportal.org/ (accessed on 30 May 2023) and modified.

**Figure 4 genes-14-01530-f004:**
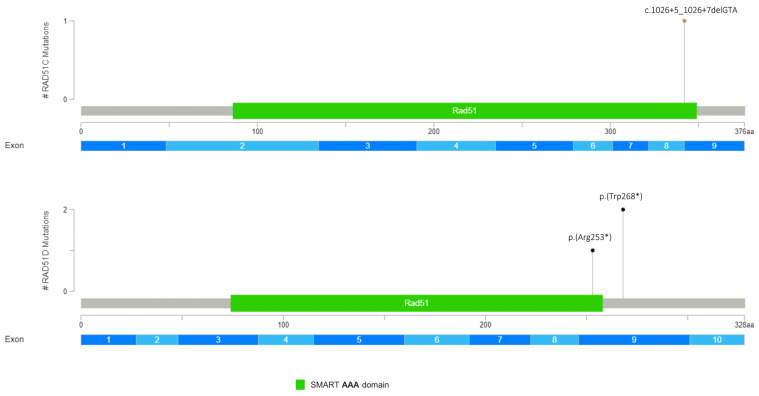
Germline PVs/LPVs identified in *RAD51C* and *RAD51D* genes. Lolliplot shows deleterious germline variants throughout the whole protein sequences of *RAD51C* (**above**) and *RAD51D* (**below**). The horizontal scale bar represents the length (amino acids) of the protein sequence, the light blue bar below represents the corresponding exons. The vertical bar represents the number of patients affected by the specific mutation (#). Each variant identified in this study is depicted by a lolliplot. In detail, the orange lolliplot identifies the splicing mutation, and the black ones display non-sense mutation. The lolliplot graph was obtained by the informatics tool Mutation Mapper (https://www.cbioportal.org/ (accessed on 30 May 2023) and modified.

**Figure 5 genes-14-01530-f005:**
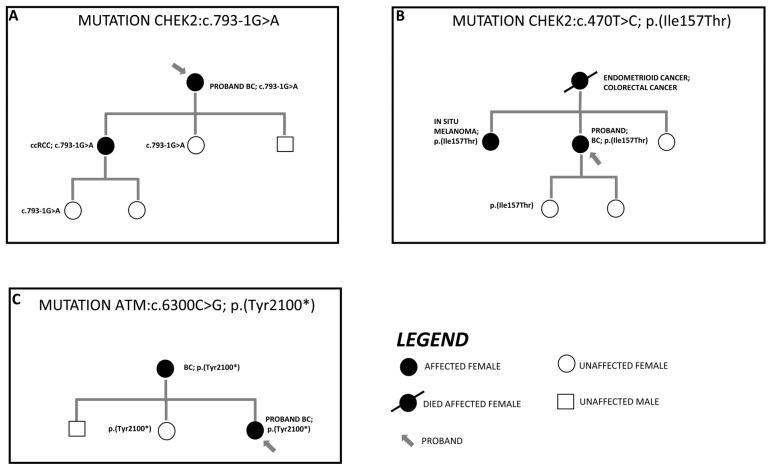
Figure illustrates three family trees showing segregation of PVs and LPVs in affected relatives. The arrows indicate the probands from each family. Panel (**A**) depicts the segregation of the *CHEK2* c.793-1G>A in a patient diagnosed with ccRCC; panel (**B**) displays the finding of the *CHEK2* p.(Ile157Thr) in an in situ melanoma case. The third family tree (**C**) illustrates the *ATM* p.(Tyr2100*) in the mother’s proband suffering from BC.

**Table 1 genes-14-01530-t001:** List of PVs and LPVs detected by multigene panel testing in BC moderate-risk genes.

	C DNA Change	Protein Change	Classification	rsID
*ATM* (NM_000051.3)	c.6300C>G	p.(Tyr2100*)	PV	rs1591789955
	c.6300C>G	p.(Tyr2100*)	PV	rs1591789955
	EXON 29 DELETION		LPV	Not applicable
	c.2638+1G>A		LPV	Not referenced
*CHEK2* (NM_007194.3)	c.1427C>T	p.(Thr476Met)	LPV	rs142763740
	c.793-1G>A		PV	rs730881687
	c.507del	p.(Phe169Leufs2*)	PV	rs587780183
	c.470T>C	p.(Ile157Thr)	LPV	rs17879961
	EXON 9 DELETION		LPV	Not applicable
*PALB2* (NM_024675.3)	c.355C>T	p.(Gln119*)	PV	Not referenced
	c.355C>T	p.(Gln119*)	PV	Not referenced
*RAD51D* (NM_002878.3)	c.803G>A	p.(Trp268*)	PV	rs750219200
	c.803G>A	p.(Trp268*)	PV	rs750219200
	c.757C>T	p.(Arg253*)	PV	rs137886232
*RAD51C* (NM_058216.2)	c.1026+5_1026+7delGTA		PV	rs587781410

**Table 2 genes-14-01530-t002:** Demographic and clinical characteristics of the selected cohort.

Characteristics	N°	%
Age at diagnosis		
21–40	4	26.7
41–60	9	60
61–80	2	13.3
Histology		
IDC-NST	11	73.3
INVASIVE MUCINOUS	1	6.7
INVASIVE LOBULAR	2	13.3
DUCTAL INTRAEPITHELIAL	1	6.7
HER2 profile		
POSITIVE	3	18.75
NEGATIVE	9	56.25
UNKNOWN	4	25
ER profile		
POSITIVE	9	56.25
NEGATIVE	5	31.25
UNKNOWN	2	12.5
PR profile		
POSITIVE	7	43.75
NEGATIVE	7	43.75
UNKNOWN	2	12.5
Family history of BC		
YES	11	73.3
NO	4	26.7
Family history of other cancer		
YES	8	53.3
NO	7	46.7

## Data Availability

The data presented in this study are available on request from the corresponding author.

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
