# Peer review of "Impact of High-to-Moderate Penetrance Genes on Genetic Testing: Looking over Breast Cancer"

_genes, 2023, doi:10.3390/genes14081530_

Round 1

Reviewer 1 Report

The authors performed germline panel genotyping of 512 hereditary breast-and ovarian cancer (HBOC) samples and provided population-specific spectrum of pathogenic variants of susceptibility genes with lower penetrance. They identified 15 pathogenic variants in five breast cancer predisposition genes beyond BRCA1/2. They embedded their results in the published literature data. Despite it is already confirmed in larger sample sets that panel tests do have benefit in HBOC, population-based screenings have relevance, because of the difference in variant frequencies among nations and divergent involvement of other susceptibility genes.

Line 56: It is not clear, that the retrospectively recruited 521 samples were consecutive HBOC patients, or selected only for those, which were genotyped earlier negative for BRCA1/2.

Line 98: You wrote, that variants were clinically evaluated by ClinVar, LOVD, Varsome and Franklin. Which was privileged and how did you judge if there was conflict among them?

Line110: It is advisable to define the MLPA probe set, for which the tests were done. It is especially important in the case of CHEK2, which is involved in more than one sets, but with different coverage completeness.

Line 131: In various populations, CHEK2 del 9 is nearly always accompanied by del 10. Did you use MLPA probe set that covered all the CHEK2 exons? Did you exclude exon 10 affection?

Table 1: It is advisable to add the registry number of the registered variants (i.e., rs17879961) and highlight the novels.

Table2: Do you have any statistical data whether clinical characteristics of the probands of deleterious variants in BC susceptibility genes other than BRCA1/2 showed any significant difference from those, harboring BRCA1/2 mutations?

Figure 5: Family member with melanoma might be only phenocopy, because CHEK2 mutations do not represent risk for this disease, let alone the c.470C>T variant, whose pathogenicity is controversial.

General remark: Gene names should be written in italics.

Reviewer 2 Report

1. The title refers to moderate-penetrance genes, so I would not mention PLAB2 as a moderate gene, it is considered moderate-high penetrance gene (life time risk 40-60%).  Breast Cancer Association Consortium. JAMA Oncol 2022; Yang X et al J Clin Oncol 2020,

2. I would not consider CHEK2 mutations classified as variants with conflicting interpretative significance with regard to pathogenicity (Tab 1), although they will probably be classified as LP or P in the near future.

3. It would be interesting to describe  intrinsic BC subtypes  (histology and immunohistochemestry) correlated with each single gene described (as mentioned in Breast Cancer Associatio Consortium JAMA Oncology 2022)

4. Authors should mention whether they detected any VUS (frequency and impact on clinical care)

Round 2

Reviewer 2 Report

Point 1, 2, 3 , 4 OK